# ALIGNING ANYTHING: HIERARCHICAL MOTION ESTIMATION FOR VIDEO FRAME INTERPOLATION

## ABSTRACT

Existing advanced video frame interpolation (VFI) methods struggle to learn accurate per-pixel motion or target-level motion. The reasons lie in that pixel-level motion estimation allows for infinite possibilities, making it challenging to guarantee fitting accuracy and global motion consistency, especially for rigid objects. Conversely, target-level motion consistency from the same moving target also breaks down when the assumption of object rigidity no longer holds. Therefore, a hierarchical motion learn scheme is imperative to promote the accuracy and stability of motion prediction. Specifically, we marry the target-level motion to the pixel-level motion to form the hierarchical motion estimation. It elaborately introduces specific semantics priors from open-world knowledge models such as the Recognize Anything Model (RAM), Grounding DIDO, and the High-Quality Segment Anything Model (HQ-SAM) to facilitate the latent target-level motion learning. In particular, a hybrid contextual feature extraction module (HCE) is employed to aggregate both pixel-wise and semantic representations, followed by the hierarchical motion and feature interactive refinement module (HIR) to simulate the current motion patterns. When integrating these adaptions to existing SOTA VFI methods, more consistent motion estimation and interpolation are predicted. Extensive experiments show that advanced VFI networks plugged with our adaptions can achieve more superior performances on various benchmark datasets.

## 1 INTRODUCTION

Video frame interpolation (VFI) aims to increase the frame rate of videos by synthesizing intermediate frames between two consecutive input frames. As a classical problem in video processing, this task has contributed to various applications, including slow-motion generation (Huang et al. (2022); Liu et al. (2024)), movie production (Siyao et al. (2021)), video compression (Wu et al. (2018)), *etc*.

Based on the granularity of motion learning, the existing technologies can be roughly divided into pixel-level and target-level technologies. The former typically predicts pixel-level motion between two consecutive input frames (**See Figure 1 (a)**), which is used to synthesize the intermediate frames by warping input frames (Liu et al. (2024)). However, pixel-level motion estimation presents infinite possibilities, which arises great difficulties for accurate motion simulation and interpolation. Though advanced techniques like global attention representation (Zhang et al. (2023); Lu et al. (2022)), are employed to refine motion estimation, the correspondence ambiguity cannot be eradicated.

Target-level technologies introduce semantic priors for efficient motion estimation (Sevilla-Lara et al. (2016); Hur & Roth (2016)). Specifically, these traditional methods divide the scene into different semantic categories, and then learn the individual motion representation. However, these methods are primarily suited for rigid motion, while the deformation or pixel-wise motion is not supported (**See deformed ball and people with non-rigid motions in Figure 1 (b)**). In addition, limited by the predefined classes, it is incapable of recognizing novel categories in the open-world.

To simulate the motion of anything in complex scenes, we explicitly introduces specific semantic priors and propose a novel hierarchical motion learning strategy for VFI. This approach seamlessly marries the target-level motion to the pixel-level motion, enhancing the accuracy and stability of motion prediction. Specifically, we elaborately introduce specific semantic priors from open-world knowledge models to facilitate latent target-level motion estimation. We first utilize Recognize Anything Model (RAM) (Zhang et al. (2024)) to tag each object in each image. Based on tagged text,

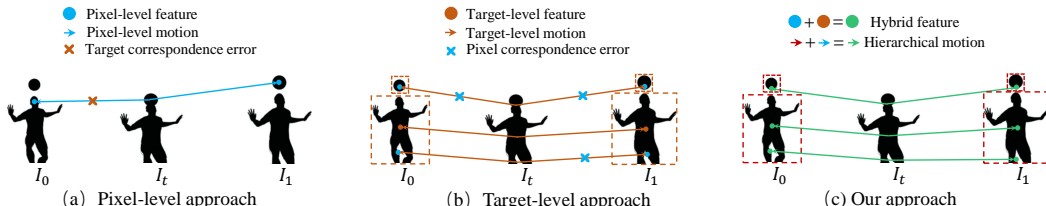

Figure 1: Different schemes for VFI. (a) Pixel-level approach: they extract pixel-level feature to predict per-pixel motion between two input frames $I_0$ and $I_1$ for intermediate frame interpolation $I_t$. (b) Target-level approach: they extract Target-level feature to predict the entire object motion for VFI. (c) Our approach: **We effectively aggregate pixel-level and target-level features to derive hybrid discriminative feature, enabling hierarchical motion estimation to simulate the current motion patterns.**

Grounding DIDO (Liu et al. (2023)) is employed to detect the corresponding objects and generate their bounding boxes. the generated bounding boxes are then used as prompts for High-Quality Segment Anything Model (HQ-SAM) (Ke et al. (2024)) to obtain specific semantic masks. Furthermore, we leverage these semantic masks as priors to enhance context extraction and motion optimization via two novel adaptations. 1) A hybrid contextual feature extraction module (HCE), which comprises of a spatial hybrid contextual feature extraction block (HCE-S) and a temporal hybrid contextual feature extraction block (HCE-T), to aggregate spatial and temporal pixel-wise and semantic representations, respectively. 2) A hierarchical motion and feature interactive refinement module (HIR), comprising a long-range hierarchical motion interactive refinement block (HIR-L) and a short-range hierarchical interactive refinement block (HIR-S), progressively simulate the motion patterns between latent intermediate frame and input frames in coarse-to-fine manner. These adaptations can be easily integrated into SOTA VFI methods. Experimental results demonstrate that SOTA methods incorporating our adaptations produce motion consistent results with minimal additional cost.

Our main contributions can be summarized as follows: 1) To the best of our knowledge, we are the first to explicitly leverage semantic information to achieve motion estimation for VFI using deep learning. 2) We propose a hybrid contextual feature extraction (HCE) to aggregate pixel-wise and semantic representation, and a hierarchical motion and feature interactive refinement module (HIR) to simulate the current motion patterns. 3) We conduct comprehensive validation of the effectiveness of our plug-and-play adaptions across a range of SOTA methods.

## 2 RELATED WORK

**Video Frame Interpolation.** The advanced VFI methods can broadly be categorized into motion-free (Kalluri et al. (2023)) and motion-based (Hu et al. (2024)) approaches, depending on whether they incorporate cues such as optical flow. **Motion-free:** This sort of method relies on phase prediction (Meyer et al. (2018)), kernel generation (Lee et al. (2020); Cheng & Chen (2021)) or spatio-temporal encoder-decoder (Choi et al. (2020); Zhang et al. (2020)) to directly produce intermediate frames. However, they lacks explicit motion modeling constraints, leading to undesirable artifacts in the interpolated results. **Motion-based:** Motion-based methods typically predict intermediate optical flows between two consecutive frames, and then leverage estimated optical flows to propagate pixels/features for intermediate frame generation (Liu et al. (2017); Jiang et al. (2018); Xu et al. (2019); Jin et al. (2023); Park et al. (2023); Liu et al. (2024)). To make VFI algorithms robust to various complex scenarios. Niklaus *et al.* extract per-pixel context information from the input frames as auxiliary information to compromise inaccuracies of optical flows (Niklaus & Liu (2018)). Bao *et al.* introduce depth information to explicitly detect occlusions, reasoning that closer pixels should be preferably synthesized in the intermediate frame (Bao et al. (2019)). Unfortunately, this sort of method focuses more on motion estimation at the pixel level, and struggles to determine the correspondences between the input frames in complex scenarios due to the lack of semantic information. Recent work has attempted to adopt SAM prior (Kirillov et al. (2023)) to explore corresponding areas in adjacent frames for better motion estimation (Han et al. (2023)). Nevertheless, they fall short in fully and explicitly utilizing semantic information, as SAM fails to identify their

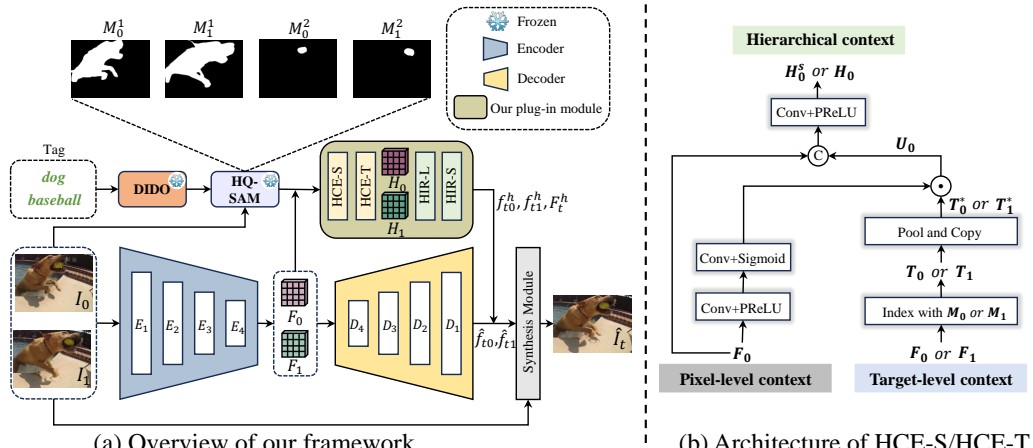

(a) Overview of our framework | (b) Architecture of HCE-S/HCE-T

Figure 2: **The overall framework (a) and model architecture (b).** Our framework consists of pixel-level baseline network (*i.e., motion estimation module and synthesis module*), pre-trained open-world models, and our plug-in module. The former extracts pixel-level feature $F_0$ and $F_1$ to predict coarse motions $\hat{f}_{t0}$ and $\hat{f}_{t1}$ and intermediate frame $\hat{I}_t$ based on two input frames $I_0$ and $I_1$, the latter two are integrated to aggregate spatial and temporal hybrid contextual features $H_0$ and $H_1$, progressively estimating long-range motions $f_{01}^h$ and $f_{10}^h$ and short-range motions $f_{t0}^h$ and $f_{t1}^h$ along with a latent intermediate feature $F_t^h$, utilizing $F_0$ and $F_1$ and generated SAM masks $M_0$ and $M_1$.

semantic classes. In this paper, we combine a pipeline that automatically tailors specific SAM masks from input frames, then these masks are used to extract target-level features and aggregate hybrid contextual feature for robust hierarchical motion estimation and frame interpolation.

## 3 METHODOLOGY

### 3.1 THEORETICAL ANALYSIS

As shown in Figure 2 (a), given two consecutive frames $I_0$ and $I_1$, pixel-level video frame interpolation (VFI) aims to predict bidirectional pixel-level motions $f_{t0}$ and $f_{t1}$ via a shared motion estimation module (ME). These motions are used to synthesize the intermediate frame $I_t$ via synthesis module (Syn). The whole process is defined as:

$$f_{t0} = ME(I_0, I_1, t), \quad f_{t1} = ME(I_1, I_0, 1-t), \quad I_t = Syn(W(I_0, f_{t0}), W(I_1, f_{t1})), \quad (1)$$

where $W(\cdot)$ denotes backward warping (Liu et al. (2017)). By observing Eq. 1, motion estimation (Hu et al. (2024)) is the most critical step in the well-established paradigms of VFI networks (Note that Syn functions as a post-processing module and is not a key focus of this paper). To analyze motion estimation comprehensively, from a probabilistic point of view, taking motion estimation in one direction as an example, the process can be expressed as:

$$\hat{f}_{t0} = \underset{f_{t0}}{\arg\max}\, p(f_{t0}|I_0, I_1, t) = \frac{p(t)p(I_0|t)p(f_{t0}|I_0,t)p(I_1|I_0,f_{t0},t)}{p(I_0,I_1,t)}, \quad (2)$$

where $\hat{f}_{t0}$ is the most likely estimated motion, and $p(f_{t0}|I_0, I_1, t)$ is the posterior distribution of the motion. we omit unrelated terms and take the logarithm to simplify the multiplication terms:

$$\hat{f}_{t0} = \underset{f_{t0}}{\arg\max}\Big\{ \underbrace{\log p(f_{t0}|I_0,t)}_{\text{context}} + \underbrace{\log p(I_1|I_0,f_{t0},t)}_{\text{interactivity}} \Big\}. \quad (3)$$

Similarly, motion estimation in the other direction can be expressed as:

$$\hat{f}_{t1} = \underset{f_{t1}}{\arg\max}\Big\{ \underbrace{\log p(f_{t1}|I_1,1-t)}_{\text{context}} + \underbrace{\log p(I_0|I_1,f_{t1},1-t)}_{\text{interactivity}} \Big\}. \quad (4)$$

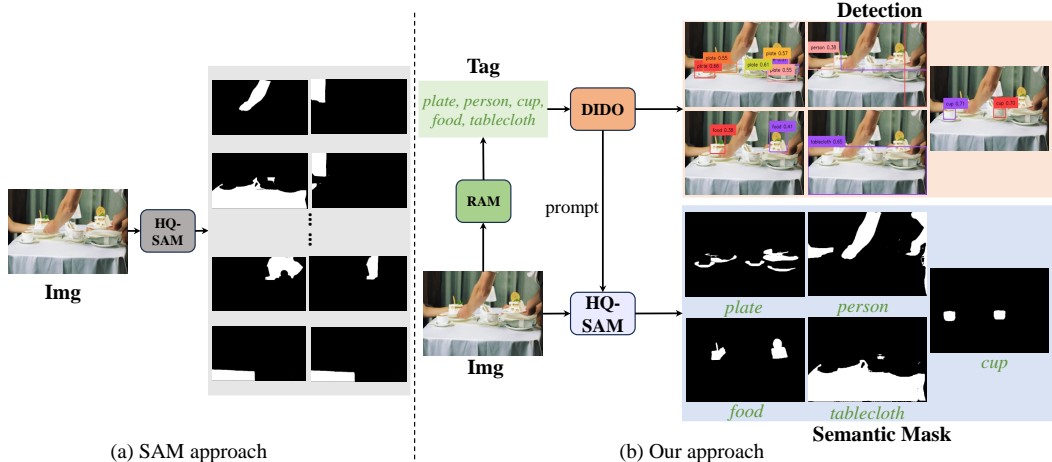

Figure 3: Different schemes for semantic mask generation. (a) SAM approach: Though SAM/HQ-SAM have shown strong capabilities in segmenting Anything, the masks generated by SAM do not specify the semantic classes and contain redundant semantic information (See the third and fourth rows). (b) Our approach: We leverage open-world knowledge models such as Recognize Anything Model (RAM), Grounding DIDO (DIDO) and High-Quality Segment Anything Model (HQ-SAM) to generate specific masks, where each pixel is basically assigned to a mask.

The context terms from Eq 3 and 4 provide an alternative information source ($I_0$ and $I_1$) for motion guidance ($\hat{f}_{t0}$ and $\hat{f}_{t1}$), respectively. However, pixel-wise contextual features extracted by existing methods (Niklaus & Liu (2018); Bao et al. (2019)) are limited to local spatial or temporal cues. In contrast, we introduce semantic priors to capture spatio-temporal local and global hybrid features as contextual information. Since intermediate frame $I_t$ is unavailable, the interactivity terms from Eq 3 and 4 highlight the interactive relationship between warped intermediate features ($W(I_0, f_{t0})$, $W(I_1, f_{t1})$) and flow ($f_{t0}, f_{t1}$), maintaining their consistency in a joint optimization manner (Kong et al. (2022); Li et al. (2023)). Unfortunately, existing methods struggle to guarantee fitting accuracy and global motion consistency since they overlook the target-level information. Unlike them, we introduce semantic priors that marry target-level motion to the pixel-level motion, achieving hierarchical motion and feature interactive refinement for current motion patterns modeling.

## 3.2 OVERVIEW

Given a pixel-level baseline network (*e.g., IFRNet (Kong et al. (2022))*) developed for VFI with target-level information, we extend it to robust motion estimation and interpolation by incorporating hierarchical information from pre-trained open-world models and our plug-in module. Specifically, as shown in Figure 2(a), given two consecutive frames $I_0$ and $I_1$, pixel-level baseline network typically employs a motion estimation module and a synthesis module to predict per-pixel motion $\hat{f}_{t0}$ and $\hat{f}_{t1}$ as well as interpolated frame $\hat{I}_t$. Our plug-in module utilizes SAM masks $M_0$ and $M_1$ from open-world models to aggregate spatial and temporal pixel-wise and semantic representations, forming hybrid contextual feature $H_0$ and $H_1$ via spatial and temporal hybrid contextual feature extraction module (HCE-S and HCE-T). These features are then fed into the long-term and short-term hierarchical motion and feature interactive refinement module (HIR-L and HIR-S) to predict hierarchical motions $f_{t0}^h$ and $f_{t1}^h$ and latent intermediate feature $F_t^h$.

## 3.3 SAM MASKS GENERATION

The powerful capabilities of SAM have showcased its versatility across various computer vision (CV) tasks (Kalluri et al. (2023); Ke et al. (2024)). In the VFI task, the key challenge lies in accurately identifying corresponding regions between input frames to improve motion estimation. Naturally, introducing semantic information enhances traditional pixel-level VFI methods by providing higher target-level representations. However, unlike semantic segmentation, as shown in Figure 3

(a), the masks generated by SAM do not specify semantic classes. Additionally, a pixel may belong to multiple different generated SAM masks (See the third and fourth rows). As a result, during training, the model struggles to simultaneously select aligned semantic information across two input frames, Moreover, they utilize redundant semantic information.

To overcome the limitation of SAM for VFI, we propose an extended SAM-based pipeline that generates specific semantic masks, ensuring that each pixel is basically assigned to a mask. As illustrated in Figure 3 (b), we begin with Recognize Anything Model (RAM) (Zhang et al. (2024)), which tags each object in the image. Based on tagged text, Grounding DIDO (Liu et al. (2023)) is introduced to detect the corresponding objects and generate their bounding boxes. these bounding boxes then serve as prompts for High-Quality Segment Anything Model (HQ-SAM) (Ke et al. (2024)) to produce specific semantic masks. To further ensure that temporal consistency and accuracy of specific semantic masks across frames, we implement the following refinement guidelines for the final tag files: 1) Discard any undetectable or irrelevant words from two tag files corresponding to two input frames. 2) Create a common tag file by taking the intersection of the two tag files, ensuring that each visible object across two input frames is associated with a corresponding semantic mask. 3) If the area of intersection between two generated masks exceeds 10% of the area of the smaller mask, we remove the word from the common tag file corresponding to the smaller mask, ensuring that each pixel is basically assigned to a mask (Note that we create a new mask to cover the remaining pixels, which are not be assigned to any mask, such as untagged classes in the tag file).

### 3.4 Our Plug-in Module

Our plug-in module is composed of two components: spatial and temporal hybrid contextual feature extraction modules (HCE-S and HCE-T), long-range and short-range hierarchical motion and feature interactive refinement modules (HIR-L and HIR-S). As analyzed in Sec.3.1, the former aggregates spatio-temporal local and global hybrid features as contextual information, the latter leverages these contexts to progressively simulate accurate motion patterns via two-stage hierarchical interactive learning.

**HCE-S and HCE-T.** HCE is designed to obtain high-quality discriminative features as contextual information for motion estimation and refinement. Previous methods (Kong et al. (2022); Li et al. (2023)) independently extract features $F_0$ and $F_1$ from a weight-sharing convolutional network, but they struggle to capture global spatial context and overlook their temporal mutual dependencies. In this paper, as shown in Figure 2 (a), we utilize generated SAM masks to index target-level features, which combine pixel-level features to model global spatial and temporal mutual relationship:

$$\begin{aligned} \textbf{Spatial global model:} \quad & H_0^s = CS(F_0, M_0), \quad H_1^s = CS(F_1, M_1), \\ \textbf{Temporal global model:} \quad & H_0 = CT(H_0^s, H_1^s, M_0, M_1), \quad H_1 = CT(H_1^s, H_0^s, M_1, M_0), \end{aligned} \quad (5)$$

where $CS(\cdot)$ and $CT(\cdot)$ correspond to HCE-S and HCE-T modules, respectively. $M_0$ and $M_1$ represent the corresponding SAM masks. $H_0^s$ and $H_1^s$ are the spatial global hybrid contextual features. $H_0$ and $H_1$ denote the spatio-temporal global hybrid contextual features. More specifically, as shown in Figure 2 (b), taking $CS(\cdot)$ to obtain $H_0^s$ as an example (Note that the mechanism of $CS(\cdot)$ and $CT(\cdot)$ is the same, only their inputs are different), the input feature $F_0$ from the encoder is separately fed into two branches, one branch maintains pixel-level local features $F_0$, while the other sequentially indexes the corresponding target-level global features $T_0^{i*}$ using generated SAM masks $M_0$ ($M_0 = \{M_0^i \mid i = 1, 2, \ldots, n\}$, where $n$ is the number of SAM masks), followed by global pooling and copy operations:

$$\textbf{Target-level global contexts :} \quad T_0^{i**} = Copy(Pool(Id(F_0, M_0^i))), \quad (6)$$

where $Id(\cdot)$ denotes index operation. $Pool(\cdot)$ and $Copy(\cdot)$ refer to average pooling and copy operations within the index regions, respectively. To further enhance feature selection and aggregation, we employ spatial attention to compute the similarities between $F_0$ and $T_0^*$ to extract useful information $U_0$, followed by merging pixel-level features $F_0$ to obtain the final hybrid contexts $H_0^s$:

$$\begin{aligned} \textbf{Selection:} \quad & U_0 = Sigmoid(Conv(PConv(F_0))) \odot T_0^*, \\ \textbf{Aggregation:} \quad & H_0^s = PConv(F_0, U_0), \end{aligned} \quad (7)$$

where $Conv(\cdot)$ and $PConv(\cdot)$ represent a convolutional layer and a convolutional layer with PReLU activation, respectively. $\odot$ denotes the element-wise multiplication.

**HIR-L and HIR-S.** HIR is designed to simu-
late more accurate motion patterns using two-
stage hierarchical motion and feature interac-
tive refinement. Traditional methods directly
predict intermediate motions at the pixel-level,
which presents challenges due to the infinite
possibilities of motion estimation, make it hard
to guarantee fitting accuracy and global motion
consistency. Moreover, predicting intermediate
motions precisely in one attempt is challeng-
ing because the intermediate frame is unavail-
able. In this paper, as shown in Figure 2 (a), we
combine target-level motion and feature with
pixel-level motion and feature for hierarchical
interactive refinement. Specifically, we perform
long-range (LR) target-level motion (Mo) and

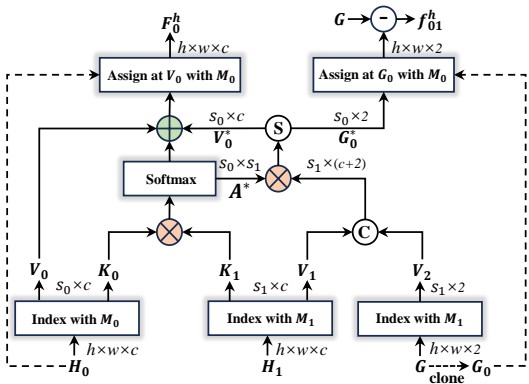

Figure 4: The architecture of HIR-L.

feature (Fe) interactive modeling, using HIR-L to estimate coarse intermediate motions and fea-
ture. Following this, short-range (SR) pixel-level motion and feature interactive modeling is used
to further predict fine intermediate motions and feature via HIR-S. This two-stage coarse-to-fine hi-
erarchical motion scheme progressively simulates more accurate intermediate motions and feature.
The whole process is expressed as:

**LR target Mo and Fe:** $F_0^h, f_{01}^h = IL(H_0, H_1, M_0, M_1), \quad F_1^h, f_{10}^h = IL(H_1, H_0, M_1, M_0).$

**Latent intermediate Mo:** $\hat{f}_{t0} = t \cdot f_{10}^h, \quad \hat{f}_{t1} = (1 - t) \cdot f_{01}^h.$

**Latent intermediate Fe:** $\hat{F}_t^h = Fuse(W(F_0^h, \hat{f}_{t0}), W(F_1^h, \hat{f}_{t1})).$ (8)

**SR pixel Mo and Fe:** $\hat{f}_{t0}^h, \hat{F}_{0t}^h = IS(\hat{F}_t^h, F_0^h), \quad \hat{f}_{t1}^h, \hat{F}_{1t}^h = IS(\hat{F}_t^h, F_1^h),$

**Interactive learning:** $F_t^h, f_{t0}^h, f_{t1}^h = Inter(\hat{F}_{0t}^h, \hat{F}_{1t}^h, \hat{f}_{t0}^h, \hat{f}_{t1}^h).$

where $IL(\cdot)$ and $IS(\cdot)$ refer to HIR-L and HIR-S. $F_0^h$ and $F_1^h$ are enhanced $H_0$ and $H_1$, $f_{01}^h$ and
$f_{10}^h$ are bidirectional LR motions between two input frames. $\hat{f}_{t0}$ and $\hat{f}_{t1}$ are linear approximations
of the bidirectional latent intermediate motions. $Fuse(\cdot)$ denotes fusion operation. $\hat{F}_t^h$ is latent
intermediate feature. $\hat{F}_{0t}^h$ and $\hat{F}_{1t}^h$ are enhanced latent intermediate features. $\hat{f}_{t0}^h$ and $\hat{f}_{t1}^h$ are enhanced
latent intermediate flows. $Inter(\cdot)$ denotes interactive refinement block (Kong et al. (2022)). More
specifically, as shown in Figure 4, taking $IL(\cdot)$ to obtain $F_0^h$ and $f_{01}^h$ as an example (Note that the
mechanism of $IL(\cdot)$ and $IS(\cdot)$ is the same, only their inputs are different), based on key-value pairs
$((K_0^i, V_0^i)$ and $(K_1^i, V_1^i)$ from $H_0$ and $H_1$ indexed by $i^{th}$ $M_0$ and $M_1$ , we computer the attention
map between them. With the global correlation matrix $A^{i*}$, we simultaneously compute the global
feature and the long-range motion from each indexed area by aggregating 1) the value $V_1^i$ of $H_1$ and
2) the value $V_2^i$ of the 2D coordinates grid $G$, respectively. the whole process is expressed as:

$$A^{i*} = \text{Softmax}\left(\frac{K_0^i K_1^{i^T}}{\sqrt{D}}\right), \quad V_0^{i*} = A^{i*}V_1^i, \quad G_0^{i*} = A^{i*}V_2^i,$$

$$F_0^{ih} = id(V_0^i + V_0^{i*}, M_0^i), \quad f_{01}^{ih} = id(G_0^{i*} - G_0^i, M_0^i).$$

(9)

Note that generated SAM masks segment the inputs into different semantic layers, allowing us to
specify target region ($S = S_1 + S_2+, ..., +S_n$) for more effective long-range motion estimation.
Moreover, the computational cost $O(S_1^2 + S_2^2+, ..., +S_n^2) < O(S^2)$ is significantly reduced as the
key matching and value retrieval are implemented as a matrix inner-product.

## 4 EXPERIMENTS

### 4.1 BENCHMARKS.

We evaluate our framework on various benchmarks containing diverse motion scenes for a com-
prehensive comparison. Structural Similarity Index (SSIM) (Wang et al. (2004)) and Peak Signal-
to-Noise Ratio (PSNR) are used as evaluation metrics. The benchmarks statistics are summarized
below:

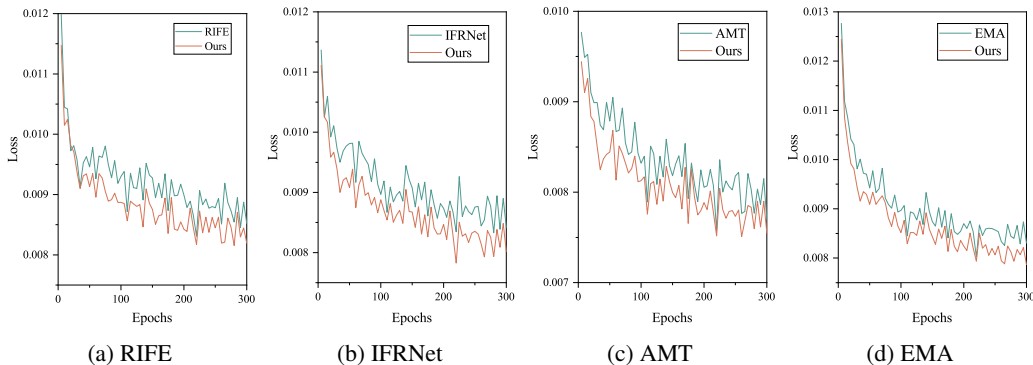

(a) RIFE  (b) IFRNet  (c) AMT  (d) EMA

Figure 5: Comparison of convergence curves for different methods integrated with our module (Note that the loss value is sampled every 5 epochs).

Table 1: Comparison of interpolation methods on different datasets and metrics. Best values (PSNR/SSIM) are highlighted in **bold**.

| Method | Type | Vimeo90K | SNU-FILM | | | | FLOPs |
|---|---|---|---|---|---|---|---|
| | Pixel-level | | Easy | Medium | Hard | Extreme | (T) |
| RIFE (Huang et al. (2022)) | CNN | **35.40**/0.9777 | **40.14/0.9908** | 35.74/0.9790 | 30.11/0.9331 | 24.81/0.8535 | 0.16 |
| RIFE_Ours | | 35.37/**0.9779** | 40.10/0.9907 | **35.80/0.9791** | **30.24/0.9346** | **25.02/0.8573** | 0.18 |
| IFRNet (Zhang et al. (2023)) | CNN | 35.52/0.9783 | **40.04/0.9905** | 35.84/0.9791 | 30.38/0.9355 | 25.09/0.8583 | 0.21 |
| IFRNet_Ours | | **35.68/0.9789** | 39.97/**0.9905** | **35.92/0.9794** | **30.48/0.9363** | **25.16/0.8599** | 0.23 |
| AMT (Li et al. (2023)) | CNN | 36.21/0.9832 | **40.01/0.9912** | 36.08/0.9805 | 30.68/0.9381 | 25.37/0.8640 | 0.66 |
| AMT_Ours | | **36.24/0.9836** | **40.01/0.9917** | **36.10/0.9808** | **30.71/0.9384** | **25.45/0.8646** | 0.69 |
| EMA (Zhang et al. (2023)) | Transformer | 35.87/0.9792 | 40.04/0.9907 | 35.82/0.9791 | 30.29/0.9346 | 25.11/0.8585 | 0.38 |
| EMA_Ours | | **35.96/0.9796** | **40.05/0.9908** | **35.93/0.9794** | **30.37/0.9350** | **25.17/0.8599** | 0.48 |

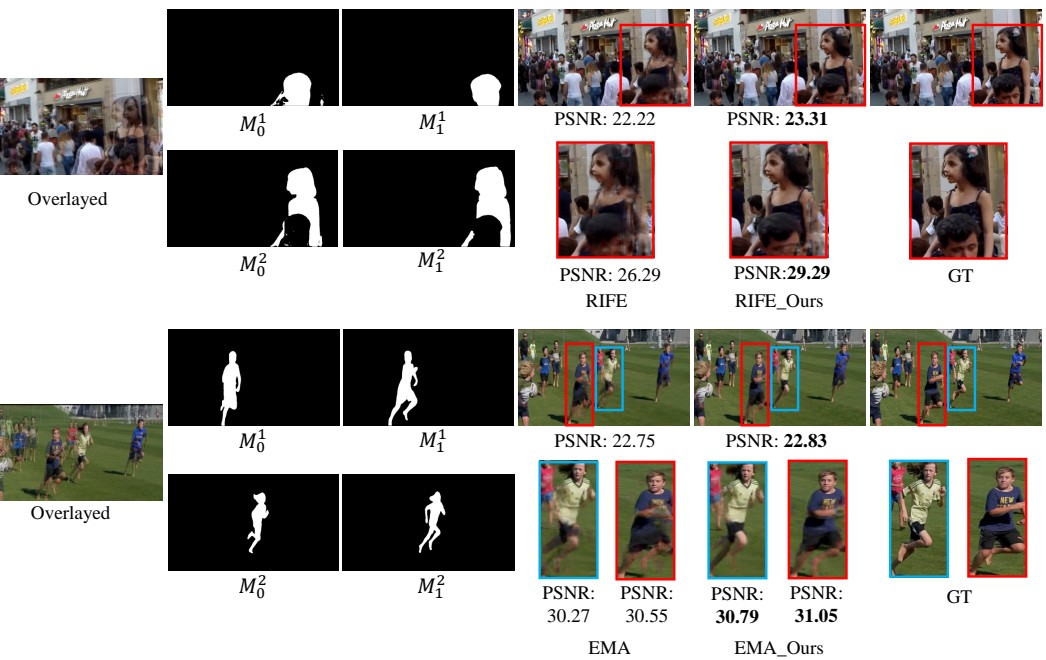

Figure 6: Visual comparisons of different VFI methods on SNU-FILM (Extreme) dataset (Note that due to space constraints, we only display a limited generated SAM masks).

**Vimeo90K (Xue et al. (2019)).** This dataset contains over 60,000 triplets with the image resolution of 448×256. A total of 51,312 triplets are cropped into patches of 224×224 pixels for training, while 3,782 triplets are reserved for testing.

**SNU-FILM (Choi et al. (2020)).** This testset includes 1,240 triplets of videos of resolution up to 1280×720, which is very challenging for large motions and occlusions scenarios. It is divided into four categories: Easy, Medium, Hard, and Extreme.

## 4.2 IMPLEMENTATION DETAILS

We integrate pre-trained open-world models and our plug-in module into SOTA methods, and train the entire model through Charbonnier loss (Charbonnier et al. (1994)) in an end-to-end manner. Specifically, we implement each model using the AdamW optimizer (Loshchilov & Hutter (2017)) through four RTX 4090 GPUs. The Vimeo90K trainset (Xue et al. (2019)) is used to train each model for 300 epochs, with a batch size of 24 and a patch size of 224×224. The learning rate is initially set to $1 \times 10^{-4}$ and gradually decays to $1 \times 10^{-5}$ following a cosine attenuation schedule.

## 4.3 COMPARISONS WITH THE SOTAS

We integrate the generated SAM masks and our plug-in module into representative SOTA methods, including RIFE (Huang et al. (2022)), IFRNet (Kong et al. (2022)), EMA-VFI (Zhang et al. (2023)) and AMT (Li et al. (2023)) for a comprehensive comparison. The computation cost of each method is measured on a 1280 × 720 resolution. To ensure a fair comparison, we retrain SOTA methods using their respective source codes, adhering to the same training strategy outlined in implementation details to train their corresponding plug-in framework.

**Quantitative Comparison. Training:** To validate the effectiveness of our strategies, we conduct a quantitative analysis during training. Figure 5 presents comparisons of convergence curves for various methods integrated with our module. The observed trends align with our theoretical analysis in Sec.3.1, demonstrating that introducing target-level information for hierarchical motion estimation can enhance convergence limits. **Testing:** As shown in Table (1), when faced with simple scenarios on Vimeo90K and SNU-FILM datasets, SOTA pixel-level methods achieve results comparable to ours. However, our plug-in model significantly outperforms these methods in challenging scenarios through hierarchical motion estimation. Specifically, our model surpasses advanced CNN-based pixel-level methods, including RIFE (Huang et al. (2022)), IFRNet (Kong et al. (2022)) and AMT (Li et al. (2023)), by clear margins of 0.21dB, 0.07dB and 0.08dB, respectively, on the Extreme subsets of SNU-FILM dataset. This superiority arises from the inherent limitations of CNN-based pixel-level methods, which struggle with infinite possibilities in motion estimation, making accurate motion simulation and interpolation highly challenging. Moreover, their limited receptive fields hinder the ability to capture large motions effectively. Additional, Transformer-based pixel-level method EMA (Zhang et al. (2023)) employs window-based attention for motion estimation in VFI. but it still suffer from a limited receptive field in dealing with large motions between corresponding targets, resulting in a performance that is xxxdB below ours. All these results highlight the effectiveness of our SAM masks and plug-in module in VFI.

**Qualitative Comparison.** The qualitative results of SOTA methods and our corresponding plug-in methods with their PSNR values on pixel-level and target-level are shown in Figure 6. It is apparent that previous VFI methods struggle to produce sharp edges of moving objects, particularly in scenarios involving large and complex motions (See the moving adult and girl). Even transformer-based method EMA (Zhang et al. (2023)) encounters similar challenges ( See a group of moving children). The underlying issue is their inability to distinguish and match target motion between input frames. In contrast, our approach comprehensively incorporates semantic information, allowing for motion estimation and interpolation specific to corresponding regions. As a result, our model accurately synthesizes content at motion boundaries and generates credible textures with fewer artifacts (Please refer to the supplementary materials for more visualization results).

## 4.4 ABLATION STUDY

This section provides comprehensive ablation studies to evaluate the impact of each component, using RIFE (Huang et al. (2022)) as the baseline. For fair comparison, all models are trained on the Vimeo90K dataset with image patches sized 224×224, for a total of 100 epochs.

**Effects of HCE.** We conducted additional experiments to validate the effectiveness of our proposed HCE across various variations. Quantitative results are shown in Table 2(a), the baseline only utilizes

Table 2: Ablation experiments of our framework on SNU-FILM (Extreme) (Choi et al. (2020)) dataset. We report the PSNR/SSIM values of these variants, and the best result is shown in bold.

| Case | HCE-S | HCE-T | Extreme |
|------|-------|-------|---------|
| Baseline | ✗ | ✗ | 24.80/0.8544 |
| HCE$_1$ | ✓ | ✗ | 24.85/0.8546 |
| HCE$_2$ | ✗ | ✓ | 24.87/0.8549 |
| Ours | ✓ | ✓ | **24.93/0.8563** |

(a) Effects of HCE.

| Case | HIR-L | HIR-S | Extreme |
|------|-------|-------|---------|
| Baseline | ✗ | ✗ | 24.66/0.8514 |
| HIR$_1$ (Target) | ✓ | ✗ | 24.80/0.8532 |
| HIR$_2$ (Pixel) | ✗ | ✓ | 24.88/0.8557 |
| HIR$_3$ (Pixel+Traget) | ✓ | ✓ | 24.88/0.8551 |
| Ours (Target+Pixel) | ✓ | ✓ | **24.93/0.8563** |

(b) Effects of HIR.

contexts from the encoder to predict hierarchical motion estimation for VFI. Building on this, HCE$_1$ and HCE$_2$ incrementally introduce spatial and temporal hybrid contextual feature extraction blocks (HCE-S and HCE-T), resulting in gains of 0.05dB and 0.07dB, respectively. This demonstrates that our HCE can effectively capture local and global contextual information. By integrating these two blocks for VFI, we achieve an even better performance improvement of 0.13dB.

**Effects of HIR.** To verify the important of our HIR in motion estimation, we perform an ablation study comparing pixel-level and target-level motion estimation strategies. As shown in Table 2(b), compared to the baseline, introducing target-level and pixel-level motion estimation strategies significantly improves performance by 0.014dB and 0.22dB, respectively. However, rearranging the sequence of motion estimations did not affect the outcome. In fact, implementing a coarse-to-fine motion estimation from the target-level to the pixel-level yields even better results.

**Effects of SAM Masks.** We extend our ablation experiments to assess the impact of masks in testing. As illustrated in Table 3, by training our model with generated masks and setting these masks to all zeros during testing, the model essentially performs global region motion estimation, yielding improved performance. Introducing the mask focuses the model on specific semantic regions, allowing more precise motion estimation with reduced computational cost (See analysis in Sec 3.4).

Table 3: Effects of SAM Masks in testing.

| Case | SAM mask | | Extreme |
|------|----------|------|---------|
| | Train | Test | |
| Baseline | ✗ | ✗ | 24.66/0.8514 |
| Case$_1$ | ✓ | ✗ | 24.89/0.8556 |
| Ours | ✓ | ✓ | **24.93/0.8563** |

## 5 CONCLUSION

This paper explicitly introduces semantic priors for video frame interpolation, effectively bringing target-level motion to pixel-level motion to enhance the accuracy and stability of motion prediction via hierarchical learning. Specifically, we utilize open-world knowledge models, such as recognize Anything Model (RAM), Grounding DIDO, and the High-Quality Segment Anything Model (HQ-SAM), to generate specific semantic masks. Additional, we propose a hybrid contextual feature extraction module (HCE) to aggregate both pixel-wise and semantic representation, alongside the hierarchical motion and feature interactive refinement module (HIR) to simulate current motion patterns. Extensive experiments demonstrate that our method plugged with these two modules surpasses SOTA methods on various benchmark datasets.

### 5.1 DISCUSSION AND LIMITATIONS.

Firstly, while SAM masks effectively distinguish different targets for motion estimation, they lacks insight into motion trajectories. Future work could explore leveraging a large language model to precisely detail each target's motion state. Secondly, our pipeline generates specific semantic masks but struggles to differentiate between instances due to their motions. The recent release of SAM2 (Ravi et al. (2024)) may provide a solution by replacing HQ-SAM. Thirdly, although hierarchical motion estimation is robust to video interpolate frame, our performance is somewhat influenced by the accuracy of SAM masks, and would greatly benefit from more advanced open-world recognition, detection and segmentation models.

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

# A  APPENDIX

## A.1  NETWORK ARCHITECTURE

As shown in Figure 7, the proposed interactive refinement block $Inter(\cdot)$ utilizes warped features ($\hat{F}_{0t}^h$ and $\hat{F}_{1t}^h$) and intermediate flows ($f_{t0}^h$ and $f_{t1}^h$) for joint optimization. For IFRNet (Kong et al. (2022)) and AMT Li et al. (2023), the final predicted results are latent intermediate flows ($\hat{f}_{t0}^h$ and $\hat{f}_{t1}^h$) and latent intermediate feature $F_t^h$ for compensation:

$$\textbf{Interactive learning:} \quad F_t^h, f_{t0}^h, f_{t1}^h = Inter(\hat{F}_{0t}^h, \hat{F}_{1t}^h, \hat{f}_{t0}^h, \hat{f}_{t1}^h). \tag{10}$$

For RIFE (Huang et al. (2022)) and EMA (Zhang et al. (2023)), the final predicted results are latent intermediate flows ($f_{t0}^h$ and $f_{t1}^h$) and mask $m_t^h$ for compensation:

$$\textbf{Interactive learning:} \quad m_t^h, f_{t0}^h, f_{t1}^h = Inter(\hat{F}_{0t}^h, \hat{F}_{1t}^h, \hat{f}_{t0}^h, \hat{f}_{t1}^h). \tag{11}$$

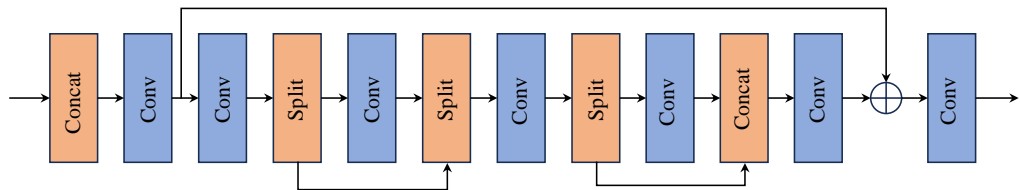

Figure 7: The architecture of interactive refinement block.

## A.2 LOSS FUNCTION

We retrain SOTA methods using their respective source codes, and only utilize Charbonnier loss (Charbonnier et al. (1994)) $\rho(x) = (x^2 + \epsilon^2)^\alpha$ ($\epsilon = 10^{-3}$) to optimize each model, denoted by:

$$L_{baseline} = \rho(\hat{I}_t - I_t), \tag{12}$$

where $\hat{I}_t$ denotes predicted result from the baseline. $I_t$ denotes ground-truth intermediate frame. For our model, in addition to supervising the final result, we also supervise the result $\tilde{I}_t$ generated by our plug-in module, and the whole loss can be expressed as:

$$L_{ours} = \rho(\hat{I}_t - I_t) + 0.1 * \rho(\tilde{I}_t - I_t),$$
$$\tilde{I}_t = m_t^h * W(I_0, f_{t0}^h) + (1 - m_t^h) * W(I_1, f_{t1}^h). \tag{13}$$

## A.3 SAM MASKS

To further ensure that temporal consistency and accuracy of specific semantic masks across frames, we implement the following refinement guidelines for the final tag files: 1) Discard any undetectable or irrelevant words from two tag files corresponding to two input frames. 2) Create a common tag file by taking the intersection of the two tag files, ensuring that each visible object across two input frames is associated with a corresponding semantic mask. 3) If the area of intersection between two generated masks exceeds 10% of the area of the smaller mask, we remove the word from the common tag file corresponding to the smaller mask, ensuring that each pixel is basically assigned to a mask (Note that we create a new mask to cover the remaining pixels, which are not be assigned to any mask, such as untagged classes in the tag file). More SAM masks visualizations are shown:

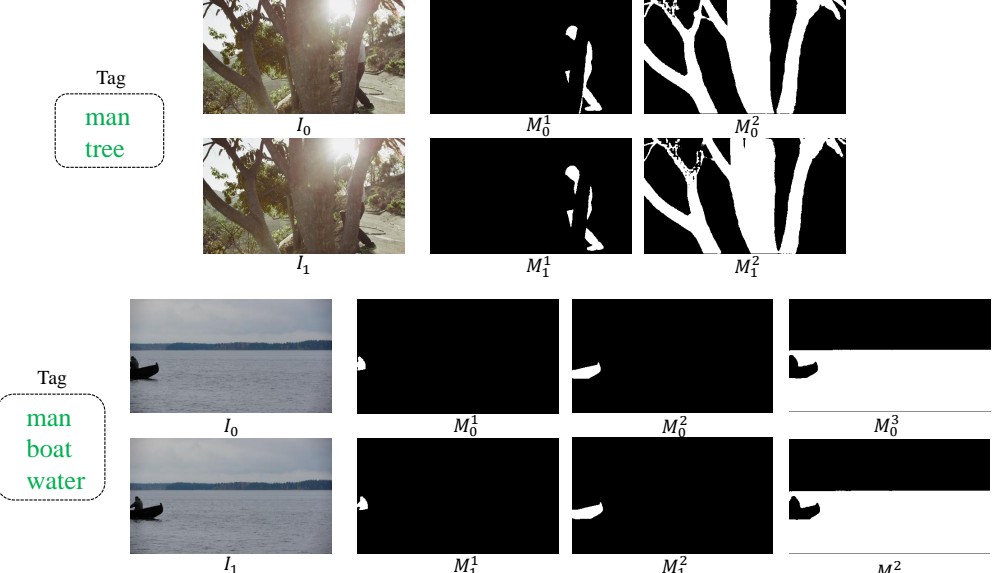

Figure 8: SAM masks visualization.

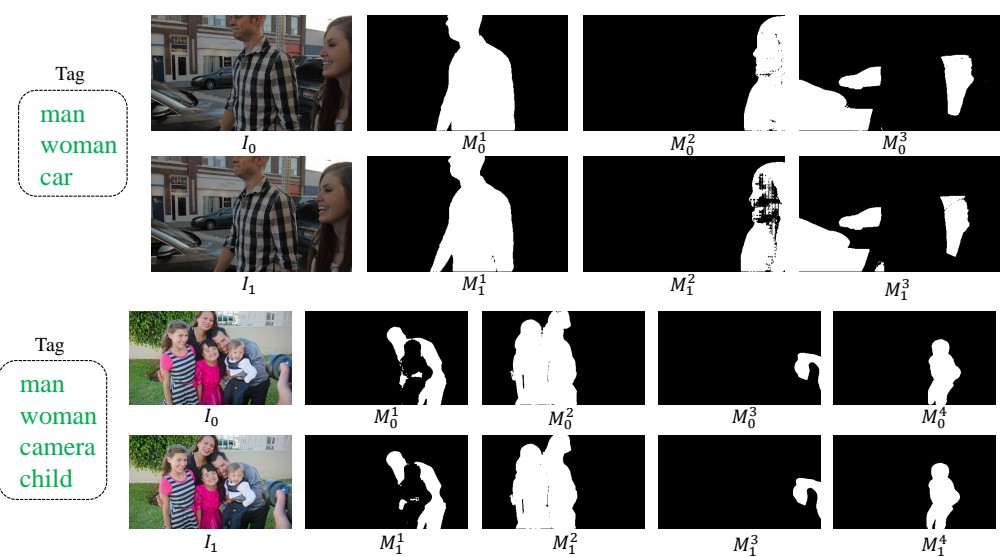

Figure 9: SAM masks visualization.

## A.4 MORE VISUAL RESULTS

In this section, we show more results of all visualizations from our plug-in network and the baseline. As shown in Figure 10, Figure 11 and Figure 12, Our models can recover the right textures with more realistic detail with clear boundary.

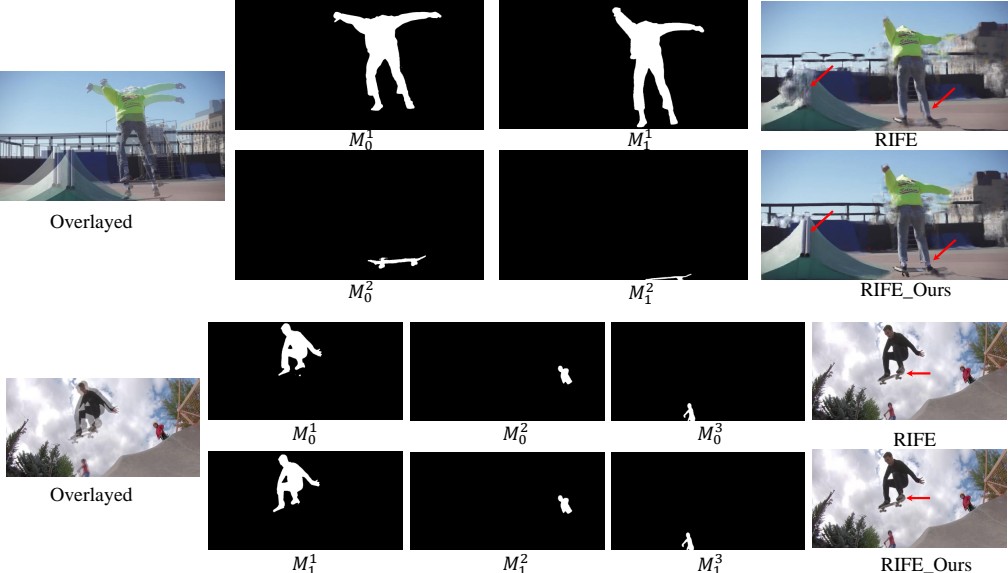

Figure 10: visual comparisons of RIFE and RIFE_Ours .

## A.5 CODE AND DEMO

We provide the completion process of our IFRNet (Kong et al. (2022)) with plug-in module in the code file, And we also provide a demo of comparison, demonstrating that our method produces more details and textures via hierarchical learning.

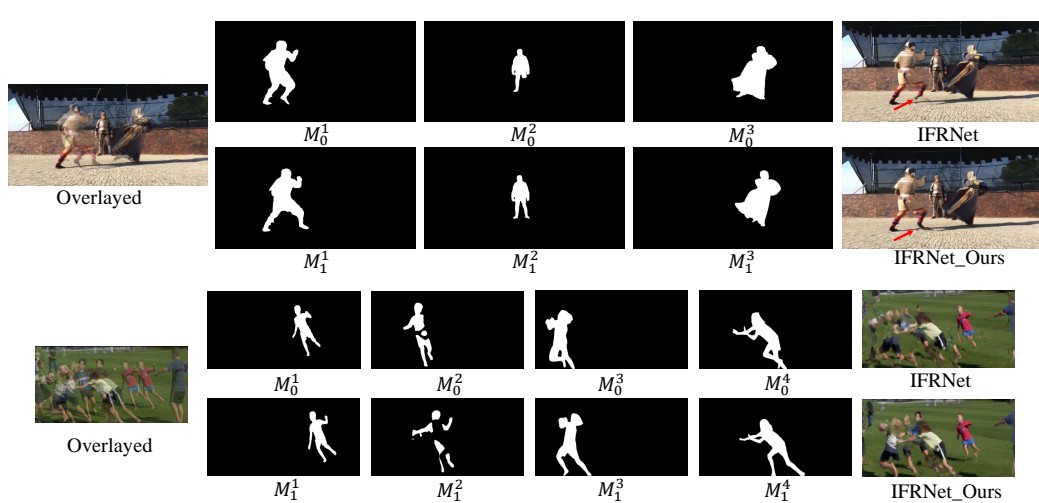

Figure 11: visual comparisons of IFRNet and IFRNet_Ours .

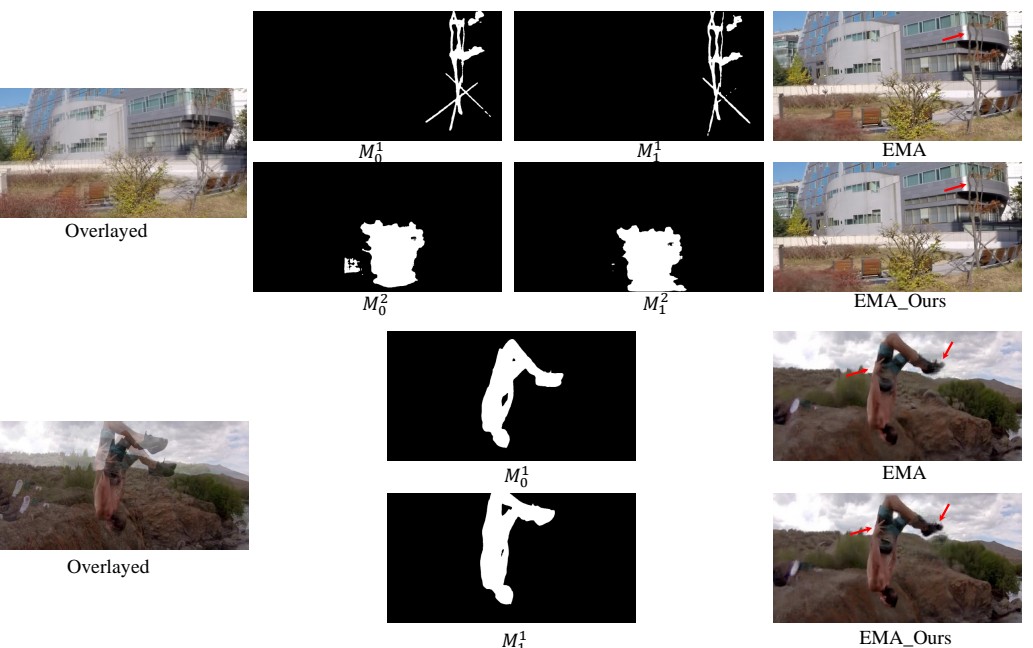

Figure 12: visual comparisons of EMA and EMA_Ours .

