# OpenReview forum: "Aligning Anything: Hierarchical Motion Estimation for Video Frame Interpolation"
_ICLR.cc/2025/Conference — ICLR 2025 Conference Withdrawn Submission_

### Official Review · Reviewer_JMs6 · 2024-11-02

**Soundness:** 2
**Presentation:** 2
**Contribution:** 2
**Rating:** 3
**Confidence:** 4

**Summary:**

The paper introduces an approach to improve video frame interpolation (VFI) by addressing challenges in motion estimation. Traditional VFI methods struggle with accurately predicting pixel-level and target-level motions due to the complexity of motion possibilities and object rigidity assumptions. The authors propose a hierarchical motion learning strategy that integrates pixel-level and target-level motions. This paper argues that semantic information and hierarchical strategies from SAM can lead to more consistent and accurate VFI results.

**Strengths:**

1.The paper attempts to solve the motion estimation problem is valuable in VFI.
2.The figure is clear.

**Weaknesses:**

1.I believe that introducing SAM to the VFI task is not valuable. Because the number of moving objects in the video cannot be determined, introducing SAM to distinguish each object will limit the generalization ability of the model and consume unstable resources. Specifically, it is recommended that the authors add experiments to demonstrate that SAM achieves the desired segmentation on VFI datasets, and that the algorithm is robust enough for different fundamental segmentation models. In addition, it is recommended that the authors analyze the frame interpolation time and resource consumption for different videos, especially for videos with only one object and those with many objects.

2.The theoretical analyses in section 3.1 are more like an introduction to preliminary work. This is because the formulas are commonly acknowledged in VFI and are not specifically linked to the authors' design. It is suggested that the authors connect video frame interpolation to the segmentation prior through theoretical analyses, in order to justify the introduction of SAM.

3. The rigid and non-rigid objects mentioned in the paper are not essentially resolved, and for non-rigid objects, SAM cannot obtain masks for intermediate frame moments. Can the authors give a detailed explanation of SAM handling non-rigid objects.

4. The mask generation process in this paper utilizes three open vocabulary models, RAM, DINO, and HQ-SAM, along with some post-processing procedures. The performance of the model is affected by them and the paper needs to be supplemented with quantitative experiments for each of them. For example, can the authors list in detail the quantitative metrics of object detection and segmentation such as mAP, mIOU, etc. to prove that DINO and SAM can cover all the objects in the video accurately enough.

5. Expanding the range of datasets used for testing could further validate the method’s robustness across different video types and scenarios. Including more diverse or challenging datasets would demonstrate broader applicability, such as XVFI, GOPRO, etc.

6. The paper could discuss potential real-world limitations or constraints, e.g., performance at frame rates X4, X8, non-uniform motion, rotation, etc.

**Questions:**

1. In this paper, all descriptions of the “Grounding DINO” are “DIDO”, including text and figure. I think this is a clerical error made by the authors during writing.

2. In Section 4.3, the authors demonstrate the performance improvement of the proposed method applied to Pixel-level SOTA models. I would like to know the effect of combining the proposed method on other types of models.

3. While the approach proposed in this paper can lead to performance improvement, the time cost of using three open vocabulary models in series needs to be considered in practical applications. It would be better if there could be relevant experiments about the inference time of the models.

---

### Official Review · Reviewer_fsWR · 2024-11-03

**Soundness:** 1
**Presentation:** 1
**Contribution:** 2
**Rating:** 3
**Confidence:** 2

**Summary:**

The paper extends existing frame interpolation methods (IFRNet, IFRNet, AMT, EMA) to introduce a module that aggregates features between two input frames. Improvement in reconstruction quality (PSNR/SSIM) is reported.

**Strengths:**

Leveraging semantics to augment motion makes sense. We can infer how dynamic or static something is from the category (e.g. more motion in the foreground).

Modularity: The "plug-in" module is shown to extend four existing architectures.

**Weaknesses:**

Writing quality: The paper was difficult to read. I noticed more language errors than in a typical submission, and the core technical contributions (e.g. how the aggregation works) should be illustrated more clearly.

Experiment suggestion: Use ground truth semantic segmentation to demonstrate the upper bound of model accuracy and how much the quality of the segmentation matters.

**Questions:**

Have you noticed any patterns in when its performance drops or when it fails significantly?

---

### Official Review · Reviewer_3ycR · 2024-11-04

**Soundness:** 3
**Presentation:** 2
**Contribution:** 2
**Rating:** 5
**Confidence:** 5

**Summary:**

The authors propose a hierarchical scheme for estimating accurate motion information for video frame interpolation, where semantic information obtained from previously introduced segmentation models are utilized. Using target-level motion information from RAM, Grounding DIDO and HQ-SAM models, the goal of the proposed method is to strike a good balance between estimating pixel-level motion & target-level motion. The authors validate their proposed method on well-known video frame interpolation datasets such as VIMEO90K and SNU-FILM, and compare their method against other state-of-the-art VFI methods.

**Strengths:**

- The motivation for the proposed method seems plausible and technically sound. Additionally, motivation of their method is straightforward and easy to understand, where finding a good balance between pixel-level and object-level motion is crucial for estimating accurate motion.

- The proposed method successfully integrates multiple models where detection and segmentation models are fused with frame interpolation models, using their proposed framework.

- The authors validate their approach with two well-known video frame interpolation datasets, and compare their algorithm with recent SoTA methods.

**Weaknesses:**

- Although the performance improvements are somewhat consistent throughout various baseline algorithms and datasets, the performane gains obtained by the proposed HCE modules seem very marginal considering the hefty additional computational load on top of the baseline algorithm.

- Since the goal of a video frame interpolation model is to synthesize the intermediate frames given consecutive set of input frames, the input frames often include fast-moving objects with varying degree of blurry artifacts. Given this nature of the VFI problem, I have two questions.

  (1) How successful do baseline detection and segmentation algorithms HQ-SAM, RAM, and DIDO perform on blurry frames? It seems that the semantic information obtained from these algorithms play a crucial role for the proposed framework, and missed detections and segments can be critical for the performance.

  (2) Considering the aforementioned aspects, using binary segmentation masks seems inappropriate given that the object boundary can be often blurry due to varying degree of motion blur. It seems that using soft masks rather than hard binary masks can alleviate this problem. Are there any experimental validations for this?


- Although using open-set models such as RAM and DIDO seems general enough for many cases and scenarios, explicitly tagging object categories, finding bounding boxes, and obtaining segmentation masks seems to limit the capability of the proposed framework to pretrained concepts. Can the proposed method utiilze semantic information from objects of unseen category?


- Minor points

  There are many typos and grammatical errors throughout the entire manuscript, and even in the abstract section. I suggest the authors to perform a full top-to-bottom proofreading of the manuscript.

**Questions:**

Please refer to the questions in the weaknesses section.

---

### Official Review · Reviewer_Ywe8 · 2024-11-08

**Soundness:** 2
**Presentation:** 3
**Contribution:** 1
**Rating:** 5
**Confidence:** 3

**Summary:**

This paper addresses the challenges in video frame interpolation (VFI) by proposing a hierarchical motion learning scheme that combines pixel-level and target-level motion estimation. It introduces semantic priors from open-world knowledge models, such as RAM, Grounding DIDO, and HQ-SAM, to enhance motion prediction accuracy and stability. A hybrid contextual feature extraction module (HCE) is designed to integrate spatial and temporal representations, while a hierarchical motion and feature interactive refinement module (HIR) simulates motion patterns in a coarse-to-fine manner. The proposed adaptations can be easily integrated into existing state-of-the-art VFI methods, leading to more consistent motion estimation. Extensive experiments demonstrate the superior performance of these enhanced VFI networks across various benchmark datasets. Key contributions include the novel use of semantic information for motion estimation and the development of the HCE and HIR modules.

**Strengths:**

1. The motivation is relatively direct, allowing readers to quickly grasp the focus of the article and gain a comprehensive understanding of the field.
2. The author's explanation of the methods is clear, enabling readers to easily understand the role of the introduced modules within the overall model.
3. The article provides a thorough and rigorous interpretation of the overall method's formulas, effectively presenting the complete model.

**Weaknesses:**

1. The proposed method in the article lacks novelty, primarily focusing on enhancing the video frame interpolation task through the utilization of foundational models such as HQ-SAM, Grounding DINO, and RAM. While it is foreseeable that the introduction of these methods could improve the performance of video frame interpolation, such enhancements are insufficient to support a highly impactful publication. I hope to see more insightful contributions from the authors in this regard.

2. The experimental section of the article does not demonstrate a clear performance gain from integrating this method into existing approaches. It is possible that achieving score improvements in the field of video frame interpolation (VFI) is inherently challenging; nonetheless, I would appreciate it if the authors could provide a comparison of other plug-and-play enhancement techniques alongside their proposed method to illustrate its relative superiority.

3. The qualitative comparison presented in Figure 6 of the article lacks necessary guidance and labeling, making it difficult for readers to discern the intended information. If the authors aim to highlight specific differences, I suggest improving the image's readability through localized zoom-ins and the addition of explanatory text.

4. The authors' experiments appear rather inconclusive, and I would like to see the distinctive advantages of leveraging semantic information for motion estimation. Specifically, what unique features emerge within the network when it is equipped with comprehensive semantic input? I hope the authors can derive more meaningful conclusions based on this foundation.

**Questions:**

1. I would like to know the impact on the overall model's runtime after introducing models such as Grounding DINO, RAM, and HQ-SAM. Although the authors present FLOPs data in Table 1, they do not provide a detailed explanation. I hope to see a more comprehensive and rigorous set of experiments and clarifications regarding the model's inference speed.

2. In line 413, there is a reference to xxxdB, which confuses me. Did the authors fail to fill in the value? I hope the authors can adopt a more rigorous approach to their paper and minimize such basic errors.

---

### Note · Authors · 2024-11-13

I have read and agree with the venue's withdrawal policy on behalf of myself and my co-authors.